# Intraoperative phrenic nerve stimulation to prevent diaphragm fiber weakness during thoracic surgery

**Guilherme Bresciani**[1], **Thomas Beaver**[2], **A. Daniel Martin**[3], **Robbert van der Pijl**[4], **Robert Mankowski**[5], **Christiaan Leeuwenburgh**[5], **Coen A. C. Ottenheijm**[4], **Tomas Martin**[2], **George Arnaoutakis**[2], **Shakeel Ahmed**[3], **Vinicius M. Mariani**[1], **Wei Xue**[6], **Barbara K. Smith**[3☯], **Leonardo F. Ferreira**[1,7,8*]

**1** Department of Applied Physiology and Kinesiology, University of Florida, Gainesville, FL, United States of America, **2** Department of Surgery, University of Florida, Gainesville, FL, United States of America, **3** Department of Physical Therapy, University of Florida, Gainesville, FL, United States of America, **4** Department of Cellular and Molecular Medicine, University of Arizona, Tucson, AZ, United States of America, **5** Department of Physiology and Aging, University of Florida, Gainesville, FL, United States of America, **6** Department of Biostatistics, University of Florida, Gainesville, FL, United States of America, **7** Department of Orthopaedic Surgery, Duke University School of Medicine, Durham, NC, United States of America, **8** Division of Physical Therapy, Duke University School of Medicine, Durham, NC, United States of America

☯ These authors contributed equally to this work.

\* leonardo.ferreira@duke.edu

## Abstract

Thoracic surgery rapidly induces weakness in human diaphragm fibers. The dysfunction is thought to arise from combined effects of the surgical procedures and inactivity. This project tested whether brief bouts of intraoperative hemidiaphragm stimulation would mitigate slow and fast fiber loss of force in the human diaphragm. We reasoned that maintenance of diaphragm activity with brief bouts of intraoperative phrenic stimulation would mitigate diaphragm fiber weakness and myofilament protein derangements caused by thoracic surgery. Nineteen adults (9 females, age 59 ± 12 years) with normal inspiratory strength or spirometry consented to participate. Unilateral phrenic twitch stimulation (twitch duration 1.5 ms, frequency 0.5 Hz, current 2x the motor threshold, max 25 mA) was applied for one minute, every 30 minutes during cardiothoracic surgery. Thirty minutes following the last stimulation bout, biopsies were obtained from the hemidiaphragms for single fiber force mechanics and quantitation of myofilament proteins (abundance and phosphorylation) and compared by a linear mixed model and paired t-test, respectively. Subjects underwent 6 ± 2 hemidiaphragm stimulations at 17 ± 6 mA, during 278 ± 68 minutes of surgery. Longer-duration surgeries were associated with a progressive decline in diaphragm fiber force (p<0.001). In slow-twitch fibers, phrenic stimulation increased absolute force (+25%, p < 0.0001), cross-sectional area (+16%, p<0.0001) and specific force (+7%, p<0.0005). Stimulation did not alter contractile function of fast-twitch fibers, calcium-sensitivity in either fiber type, and abundance and phosphorylation of myofilament proteins. In adults without preoperative weakness or lung dysfunction, unilateral phrenic stimulation mitigated diaphragm slow fiber weakness caused by thoracic surgery, but had no effect on myofilament protein abundance or phosphorylation.

**Data availability statement:** All relevant data are within the paper and its Supporting Information files.

**Funding:** National Institutes of Health R01 AR072328, HL130318, HL121500 The funders had no role in the study design, data collection and analysis, decision to publish, or preparation of the manuscript.

**Competing interests:** The authors have declared that no competing interests exist.

## Introduction

Thoracic surgery causes rapid (within two hours) diaphragm muscle fiber weakness accompanied by degradation of sarcomeric proteins [1]. The diaphragm is the primary inspiratory muscle and several reports suggest that inspiratory muscle weakness is a critical determinant of post-operative pulmonary complications, time on mechanical ventilation, and hospital length of stay [2–5]. Muscle fiber weakness during thoracic surgery is selective to the diaphragm as it was not evident in the latissimus dorsi [1]. The etiology and mechanism of diaphragm weakness during thoracic surgery remains largely unknown and, despite the potential post-surgery respiratory complication derived from it, there have been no efforts to treat the problem. Prolonged inspiratory unloading and diaphragm inactivity during mechanical ventilation, even without thoracic surgery, causes diaphragm fiber weakness and sarcomeric protein degradation [6–8]. The selective impact of thoracic surgery on the diaphragm suggests that inactivity per se, and not systemic factors associated with the surgical procedure (e.g., anesthetics and inflammation), is a primary cause of weakness [1]. It is worth noting that thoracic surgery, particularly an open-chest procedure, is a unique scenario where the diaphragm becomes fully quiescent because inflation and deflation of the lungs is dissociated from cyclic passive diaphragm movement that occurs during controlled mechanical with a closed chest.

Currently, there are no therapies to prevent diaphragm weakness during thoracic surgery. Preoperative inspiratory muscle training has been proposed as a method to increase diaphragm force, minimize the negative impact of surgery on the muscle, and improve post-operative outcomes [2,9]. The benefit of preoperative inspiratory muscle training for patients with inspiratory muscle weakness is well-established [2], but the evidence seems inconclusive when a broad cohort of patients is considered [9] suggesting that new approaches are needed to prevent or minimize inspiratory muscle weakness and postoperative complications in a wide group of patients undergoing thoracic surgery. Importantly, *preoperative* interventions do not directly target the inactivity that occurs *during* thoracic surgery and likely contributes to the limited benefits of preoperative interventions targeting the diaphragm.

Diaphragm activation during mechanical ventilation and thoracic surgery is possible via electrical stimulation of the phrenic nerve. Phrenic stimulation to maintain diaphragm activity during non-surgical mechanical ventilation attenuates diaphragm fiber alterations (atrophy or contractile dysfunction) in animals [10,11] and humans [12,13]. In a small pilot study, we reported that intermittent intraoperative phrenic stimulation during open-chest thoracic surgery increased diaphragm fiber force in a cohort of patients with abnormal pulmonary function tests and inspiratory muscle weakness [14]. However, the relevance of our findings to patients with preoperative pulmonary function or inspiratory muscle strength within the normal range remains unknown. In our previous study, we were also unable to examine the impact of stimulation on slow type fibers that constitute 50–60% of the human diaphragm.

The objective of the current study was to explore whether brief bouts of intermittent electrical phrenic nerve stimulation attenuate diaphragm slow and fast fiber weakness and the accompanying activation of protein degradation pathways, attenuation of protein synthesis pathways, and sarcomeric protein degradation during open-chest thoracic surgery performed in patients with a normal range of preoperative pulmonary function tests or maximal inspiratory pressure.

## METHODS

### Trial design

This was a controlled, prospective cohort study of intermittent, intraoperative phrenic nerve stimulation on diaphragm single fiber contractility. The protocol included unilateral phrenic

stimulation with the contralateral side serving as an internal control. This design benefits from intra-subject comparisons but may underestimate the protections elicited by stimulation due to stretching of the contra-lateral diaphragm (see *Limitations*)

## Patients

Eligibility criteria for intraoperative hemidiaphragm stimulation included adults (18–80 years) scheduled for non-emergent clinical cardiothoracic surgeries anticipated to last at least 3 hours, enabling 4 or more bouts of phrenic stimulation. Patients were ineligible to enter the overall study for any of the following: (1) history of surgery to the diaphragm or pleura, (2) severe obstructive lung disease (FEV1 < 40% predicted), (3) restrictive lung disease, (4) severe chronic heart failure (NYHA class IV), (5) chronic kidney disease with serum creatinine > 1.6 mg/dL, (6) body mass index < 20 or > 40 kg/m$^2$, (7) chronic uncontrolled metabolic disease, (8) concurrent neoplastic or myopathic illness, or (9) use of immunosuppressants, corticosteroids, or aminoglycoside antibiotics within 28 days of surgery. The protocol was expanded in Year 2 to allow enrollment of patients undergoing lung transplantation, but we included in the analysis reported here only data from patients from the original protocol with inspiratory muscle strength and lung function within the normal range (MIP or FEV1 and FVC > 70% predicted). The protocol was approved by the University of Florida Institutional Review Board, prospectively registered (NCT03303040), and will be posted on ClinicalTrials.gov. Patients were recruited from the University of Florida cardiothoracic surgery clinic, and each patient provided their written informed consent in advance of participation.

## Pulmonary function testing

Subjects completed forced vital capacity (FVC) maneuvers to screen for significant obstructive disease. Seated spirometry (Discovery 2, FutureMed) was conducted in accordance with recommended guidelines [15]. Additionally, maximal inspiratory pressure (PImax) was measured with a commercial manometer (MicroRPM, Vyaire Medical) using recommended procedures [16]. For both tests, trials were repeated 3–5 times, and the highest effort was reported.

## Intraoperative phrenic stimulation

All subjects underwent a midline sternotomy. During surgery, the right or left phrenic nerve was stimulated using an external cardiac pacemaker (Model #5388, Medtronic) and temporary pacing wires. Selection of the stimulated side was determined by the surgeon, based on ease of access to the corresponding phrenic nerve and the type of procedure being performed. The surgeon placed electrodes adjacent (~5–10 mm) to the phrenic nerve, either near the pulmonary artery or at the right pericardium, and confirmed the stimulating electrodes did not directly contact the phrenic nerve. Twitch stimulation parameters included a 1.5 ms pulse width and contraction frequency of 0.5 Hz (one twitch every 2 seconds) for a total of 1 min duration. Atrial output was 0 mA, and ventricular output was set at twice the visible twitch threshold (up to the device maximum current of 25mA), which ensured a vigorous hemidiaphragm contraction. We delivered the first stimulation from the time the phrenic nerves were initially exposed and repeated the protocol every 30 min until completion of the procedure. We opted for this protocol to minimize patient movement with stronger tetanic contractions, bilateral impact of stimulations, and interruption of the surgical procedure. It is worth noting that the external cardiac pacemaker used in our study is not labeled for phrenic stimulation.

## Diaphragm biopsies

Approximately 30 minutes following the final stimulation bout, diaphragm biopsies (~150 mg) were obtained from the stimulated and unstimulated costal hemidiaphragms and and immediately placed in chilled Buffer X (2.77 mM $CaK_2EGTA$, 7.23 mM $K_2EGTA$, 5.77 mM $Na_2ATP$, 6.56 mM $MgCl_2 \cdot 6H_2O$, 20 mM taurine, 15 mM $Na_2Phosphocreatine$, 20 mM imidazole, 0.5 mM dithiothreitol and 50 mM K-MES, 35 mM KCl; pH adjusted to 7.1 using 5 N KOH; 295 mosmol/kg $H_2O$), debrided of extramuscular fat, divided and weighed. Samples specified for single fiber contractile studies were preserved in ice-cold relaxing solution (in mM: 100 KCl, 20 imidazole, 4 ATP, 2 EGTA, 7 MgCl; pH adjusted to 7.0 using KOH), while tissue for titin studies were flash-frozen in liquid nitrogen until further analysis. Separate, blinded investigators conducted all of the muscle fiber analyses.

## Permeabilized single fiber mechanics

We prepared samples for permeabilized single fiber mechanics using a technique similar to that described by Campbell and Moss [17], and adapted in our lab for diaphragm muscle [18–20] with a few modifications for this study. Samples were placed in ice-cold relaxing solution and mini bundles ( ~ 4mm x 1mm) were carefully cut with fine spring scissors to avoid fiber overstretching. Mini-bundles were transferred to relaxing solution with 1% Triton X-100 (skinning solution) for 4 hours at 4°C to achieve chemical permeabilization. Bundles were then transferred to relaxing solution containing 50% glycerol (v/v) for storage at −20 °C and used within 3 weeks of sample collection to to minimize potential deterioration in function [17]. We aimed to obtain a full set of measurements in at least 6 fibers per side, but tested as many fibers as possible, within the 3-week window post-biopsy. On the day of the experiment, single fibers were isolated from permeabilized bundles in ice-cold relax solution, then mounted between a force transducer (403B, Aurora Scientific, Ontario, Canada) and a motor arm (312B, Aurora Scientific) in relaxing solution at 15 °C, and stretched to reach a sarcomere length (SL) of ~ 2.60 μm. Fiber length, width, and height were measured using video microscopy. At this point, we discarded fibers with SL > 2.4 μm under slack condition, a sign of damage imposed during fiber isolation in our hands. Fibers with slack SL ≤ 2.40 μm were then transferred to pCa 9.0 (pCa = - $\log_{10}[Ca^{2+}]$) and allowed ≥ 3 minutes for equilibration. Calcium-activated force was elicited by immersing the fiber in different pCa solutions at 15°C (in mM: 20 imidazole, 14.5 creatine phosphate, 7 EGTA, 4 MgATP, 1 free $Mg^{2+}$, and free $Ca^{2+}$ ranging from 1 nM [pCa 9.0] to 32 μM [pCa 4.5]) with sufficient KOH (semiconductor grade, Sigma Aldrich) to adjust the ionic strength to 180 mM at pH 7.0. Once force plateaued at each pCa, we performed a quick-release step by rapidly moving the motor arm toward the force transducer by a distance equivalent to 20% of the fiber length, holding for 20 ms, then returning the motor to its original position restoring the initial fiber length. Exposures to each pCa < 9.0 were interspersed by a 3-min period at pCa 9.0. This approach minimizes fiber rundown in our hands. Immediately after the last calcium activation, we completed strontium-based activation according to Hvid et al [19] with slight modifications. Briefly, two stock solutions were prepared 1) SrNF (in mM: 90 HEPES, 50 EGTA, 8.5 MgO, 40 $SrCO_3$, 8 $Na_2ATP$ and 10 $Na_2CrP$, pH 7.10 with KOH) and 2) INxF (in mM: 90 HEPES, 50 EGTA, 10.3 MgO, 8 $Na_2ATP$ and 10 $Na_2CrP$, pH 7.10 with KOH), both with free $[Mg^{2+}]$ of 1mM. Solutions INF and SrNF were carefully mixed to yield different pSr concentrations ($-\log_{10}[Sr^{2+}]$): high strontium concentration (pSr 4.0) and intermediate strontium concentration (pSr 5.0). Strontium activation followed the same procedure as described above for calcium activation. We used the ratio between force at pSr 4.0 and 5.0 to define slow and fast myosin heavy chain isoforms, where a ratio of force at pSr 5.0 to pSr 4.0 ≥ 0.75 indicates slow fibers (slow MyHC) and ratio < 0.75

indicates fast fibers (fast MyHC). Following pSr activation, single fibers were placed individually in dry Eppendorf tubes and stored at −80 °C until sample preparation for MyHC isoform analyzes through SDS-PAGE gel electrophoresis. In 10 subjects, we identified 171 slow fibers and 160 fast fibers using pSr. Two bands were identified in 13 fibers (6 from the stimulated side, 7 unstimulated) from 5 subjects. No bands were detected in 1 fiber. We compared the pSr criteria with MyHC separation by electrophoresis for definition of isoforms and found > 95% agreement between methods (Fig S1 and *Supplemental file*). We were unable to obtain pSr results for 5 slow fibers, which were classified using electrophoresis. In four patients, single fibers were classified as slow or fast based solely on the pSr data.

Experiments were performed and analyzed using SLControl software [21]. We used fiber width and height to determine fiber cross-sectional area assuming an elliptical shape. The force–pCa relationship of individual fibers was analyzed using a four-parameter Hill equation (Prism 5.0b, GraphPad Software, La Jolla, CA): $F = F_{pas} + F_o(10^{-pCa})^{nH}/[(10^{-pCa})^{nH} + (10^{-pCa50})^{nH}]$, where $F_{pas}$ is passive force, $F_o$ is maximal active force, $n$H is the Hill coefficient, and $pCa_{50}$ is the pCa that elicits half-maximal activation. The force response to the quick-release procedure was fitted using a single exponential equation to determine the rate of tension redevelopment (ktr). All fibers maintained force at pCa 4.5 ≥ 10% from the first to the last calcium activation.

## SDS-PAGE Analysis of MyHC Isoform

Single fibers were carefully placed at the bottom of each Eppendorf using a micropipette tip under a microscope. Myosin heavy chain molecules were extracted from each fiber segment by adding 15μl of MyHC extraction buffer containing (in mM) 100 KCl, 100 $KH_2PO_4$, 50 $K_2HPO_4$, 10 EDTA, 16.8 $Na_4P_2O_7$, 4 mM β-mercaptoethanol, 0.5% protease inhibitor cocktail (v/v, P8340, Sigma Aldrich) and pH 6.5 with KOH with 5% Triton X-100 (v/v) according to Tikunov et al [22]. Fibers were thoroughly agitated and incubated in shaker for 24 hours at 4°C. After, the final volume was doubled by adding 2x Laemmli Buffer (Bio-Rad) with DTT (0.35M) and samples were heated for 4 min at 95–100 °C. At this point, fibers were either directly loaded on gels or stored at −80 °C until electrophoresis was performed.

For intact, non-permeabilized whole muscle, frozen diaphragm tissue (−80 °C) was used. Approximately 1–2 mg of diaphragm sample were placed immediately in MyHC extraction buffer with 10% Triton X-100 (1:1) and gently ground in ice (1:100 w/v). Samples were then placed in a shaker and incubated for 24h at 4 °C. The following day, 20 μl of the supernatant were placed into a new Eppendorf containing 80 μl of 2x Laemmli buffer (Bio-Rad) with dithiothreitol (DTT; 0.35M) and thoroughly agitated. After adding 30 μl of glycerol per tube, samples were heated (95–100 °C) for 4 minutes and then stored at −80 °C until electrophoresis.

We determined the MyHC isoform of single muscle fiber segments and portions of whole tissue from the biopsies. The content of fast and slow MyHC isoforms in single fibers and whole diaphragm samples was determined as previously described by Chung et al [23] with a few modifications. Briefly, the modified version of the protocol consisted in preparing gels in 13.3 x 8.7 cm (W x L) 1.0 mm thick gel casting sets (3459903, Bio-Rad, Hercules, CA). Gels were run at 4 mA (Owl™ EC1000XL, Thermo Scientific, Hempstead, UK) for 36 hours at 4 °C. A diaphragm sample obtained from a Wistar rat was loaded in each gel as an internal standard to assure sample preparation and SDS-PAGE conditions allowed the correct identification of all MyHC isoforms in both single fibers and whole diaphragm samples. Gels were stained using a commercial kit (Silver Stain Plus, Bio-Rad, Hercules, CA) and scanned (Gel Doc EZ Imager; Bio-Rad) for analysis. For single fiber analyzes, we defined fibers as "slow-" or "fast-twitch" and excluded fibers that co-expressed both fast and slow isoforms. For whole diaphragm samples the relative content of each isoform was determined by fitting

the densitometry profiles with asymmetric Lorentzian functions (GelBandFitter software) as previously described [24].

## SDS gel electrophoresis and Western blotting for Titin Content

SDS-PAGE gel electrophoresis and Western blot experiments have been previously described [25]. Tissue was ground to a fine powder using Dounce-style homogenizers cooled in liquid nitrogen. Tissue powder was resuspended in a 1:1 mixture of an 8 M Urea buffer (in M; 8 urea, 2 thiourea, 0.05 Tris–HCl, 0.075 DTT, as well as 3% SDS and 0.03% bromophenol blue, pH 6.8) and 50% glycerol containing protease inhibitors (0.04 mM E-64, 0.16 mM leupeptin, and 0.2 mM PMSF). The solutions were mixed for 4 minutes, followed by 10 minutes of incubation at 60°C. Samples were centrifuged at 12.000 rpm and the supernatant was divided into smaller aliquots and flash frozen for storage at −80°C. SDS-PAGE was performed using 1% agarose gels run in a Hoefer SE600X vertical gel system (Hoefer Inc), to separate titin from other proteins. Gels were run at 15 mA per gel for 3 hours. For analysis of titin and MyHC levels gels were stained using Neuhoff's Coomassie brilliant blue staining protocol. Total phosphorylation was analyzed by using Pro-Q Diamond Gel Stain and SYPRO Ruby (Thermo Fisher) for total protein. After staining, gels were scanned using a commercial scanner Gbox (Syngen).

Titin binding proteins MARP1 (ANKRD-1; 1:1000 Myomedix), MARP2 (ANKRD-2; 1:2000 Myomedix) and CAPN3 (CAPN-3; 1:1000 Myomedix), and the stress response proteins CSRP3 (MLP-1; 1:5000 Myomedix) and vinculin (ab18058; 1:2000 Abcam) were quantified using western blot. All proteins were normalized to GAPDH (#2118; 1:5000 Cell Signaling or MA5-15738 1:4000 Pierce). To quantify titin degradation, Western blot was performed using titin N-terminal (H00007273-M06; 1:2000 Abnova) and C-terminal (TTN-9; 1:2000 Myomedix) antibodies. MyHC levels were determined from Coomassie stained initial gels to determine equalized loading for each sample and run on 4–12% acrylamide gels [25]. Proteins were transferred onto Immobilon-P PVDF 0.45 μm membranes (Millipore) using semi dry transfer (Bio-Rad) and specific regions of the membrane were cut and used to probe for proteins with distinct molecular weights or multiple channels for two proteins in the same region, which allowed us to test multiple relevant proteins with limited samples available. Membranes were blocked with Odyssey blocking buffer (Li-Cor Biosciences) for 1 hour, and subsequently probed with primary antibodies at 4°C overnight. Near Infra-Red dyes were used as secondary antibodies for dual color detection with Odyssey CLx Imaging System (Li-Cor Biosciences).

## Statistical analyses

The sample size for the trial was determined in advance of the study, based on our prior work examining intraoperative changes in diaphragm mitochondrial function [26]. We did not have specific force data from type I and II fibers in this population to perform sample size calculations for the study. We proceeded to explore effects on single fiber function under the assumption that by measuring at least 6 fibers per side, but aiming to measure as many fibers as possible, the number of subjects included would be sufficient to detect specific force differences between sides based on published studies from our group testing human fiber contractile function, e.g., [1,6,27]. Demographic characteristics of the patients were summarized using n (%) and Mean ± SD. Single fiber outcomes of interest (area, absolute force, passive tension, maximal specific force, Ktr, H and KD) were summarized with Mean, SD, and range. Then, we summarized the average measures by side and fiber type. The individual fiber data were used to compare effects of stimulation on contractile function via a generalized linear mixed model (SAS 9.4), which accounted for the repeated and correlated measures contained

with each subject's paired stimulated and control samples. The primary exploratory outcome of interest in the study was single fiber specific force. The linear mixed model approach provides superior statistical power and confidence to grouping the data and analyzing with simpler approaches, such as t-tests or ANOVA. Fixed effects included stimulation and fiber type, while the subject was treated as a random intercept effect. We confirmed homoscedasticity and normality by visual inspection of residuals. The assumption of randomness is based on inclusion of all patients that met criteria and consented to participate in the study. Relationships between fiber CSA, specific force, and duration of MV were evaluated with standard linear regression, under the assumption of linearity between variables, homoscedatiscity (confirmed by visual inspection), and independence of observations (true per experimental design). As each patient can have multiple (repeated) observations on each side for each fiber type, the average of the repeated measures for each patient on each side for each fiber type was first calculated to establish relationships with MV duration. To evaluate the abundance of titin, titin degradation products, and titin-binding proteins, the normality of each dataset was assessed with a Shapiro-Wilk test, and differences between the stimulated and unstimulated sides were compared with Wilcoxon or t-tests. We considered statistical significance when $p < 0.05$ and did not correct for multiple tests of secondary endpoints because these were performed to gain insights into potential biophysical and physiological mechanisms underlying the primary endpoint of specific force. In this context, $p < 0.05$ is meant to represent potential physiological or biophysical relevance and not clinical significance.

## Results

### Patient characteristics

Fig 1 outlines the subject enrollment information. Patients were enrolled between February 2018 through November 2020, when full enrollment was reached. Of 1,018 surgical clinic patients reviewed for eligibility, 25 subjects were enrolled. Three subjects were enrolled but did not receive intraoperative stimulation, due to preoperative changes in their clinical care plan. One subject initiated but did not complete intraoperative stimulation due to the requirement of high-dose neuromuscular blockade as part of surgical care. Of 21 patients who completed intraoperative hemidiaphragm stimulation, single fiber mechanics were not evaluated in two patients, due to unavailability of study personnel (n = 1), and secondary to non-cardiac surgery (n = 1). Table 1 illustrates the demographic characteristics of the sample (n = 19, 9 female). Patients received neuromuscular blockade agents (rocuronion or vecuronion; 50–100 mg) for induction of general anesthesia. A subset of patients (n = 5) received a maintenance dose of either agent during surgery. One patient received Sugammadex for reversal of neuromuscular block. The doses of neuromuscular blocking agents did not interfere with phrenic nerve stimulation. Patients underwent an average of 6 ± 2 intraoperative stimulations with stimulation amplitude of 17 ± 6 mA, and they were mechanically ventilated for 284 ± 62 minutes prior to biopsy. The use of neuromuscular blockade maintenance had no statistically significant effect on stimulation amplitude (maintenance NMBA YES: median 12.6 mA, range 6.8–25 mA; NO: median 20.1, range 10–25 mA; Mann-Whitney test p = 0.155). Of the 38 biopsies obtained for this study, two specimens (subject 15: unstimulated hemidiaphragm; subject 17: stimulated hemidiaphragm) contained primarily fat and connective tissue and were not analyzed (S1-S2 Table).

### Diaphragm single fiber

In total, 745 diaphragm fibers were activated. Of those, 382 were classified as slow (186 stimulated and 196 unstimulated; range 6–28 fibers per side for each subject) and 363 fibers were

## CONSORT Flow Diagram

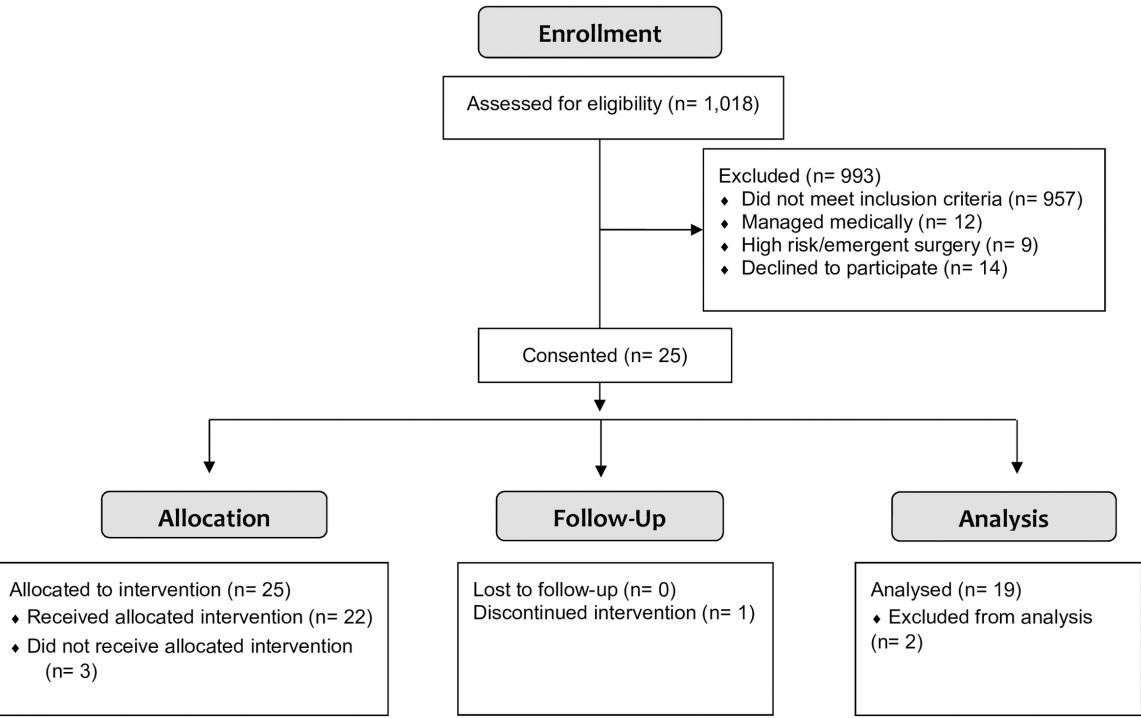

**Fig 1. Consort Flow Diagram of patient enrollment, screening, and randomization.**

**Table 1. Patient characteristics.**

|  | Mean (SD) |
|---|---|
| Age (years) | 59 (12) |
| Gender | 9F/10M |
| BMI (Kg/m²) | 30 (4) |
| %FVC | 88 (12) |
| %FEV1 | 85 (13) |
| PI Max (cmH₂O) | 90 (23) |
| MyHC1 (%) | 64 (10) |
| Time (min) Intubation to biopsy | 278 (68) |

F: female; M: male; BMI: body mass index; %FVC: forced vital capacity; %FEV1: forced expiratory volume in the first second; PImax: Maximal inspiratory pressure; CABG: coronary artery bypass grafting; AVR: aortic valve replacement; MVR: mitral valve replacement; MyHC1: type 1 myosin heavy chain; v: number of vessels grafted

classified as fast (186 stimulated and 177 unstimulated; range 7–18 fibers per side for each subject) (Fig S1). Figs 2, and S2–S4 show the effects of stimulation on single fiber size and contractile function. In slow fibers, absolute force (Fig 2A), cross-sectional area (Fig 2B), and specific force (Fig 2C) were significantly higher (10–25%) for the stimulated hemidiaphragm (all p < 0.005). We confirmed the statistical difference in the primary endpoint (specific force) with a Mann-Whitney test (p = 0.0290; median: control 91 kN/m², STIM 101 kN/m²). Fig 2D

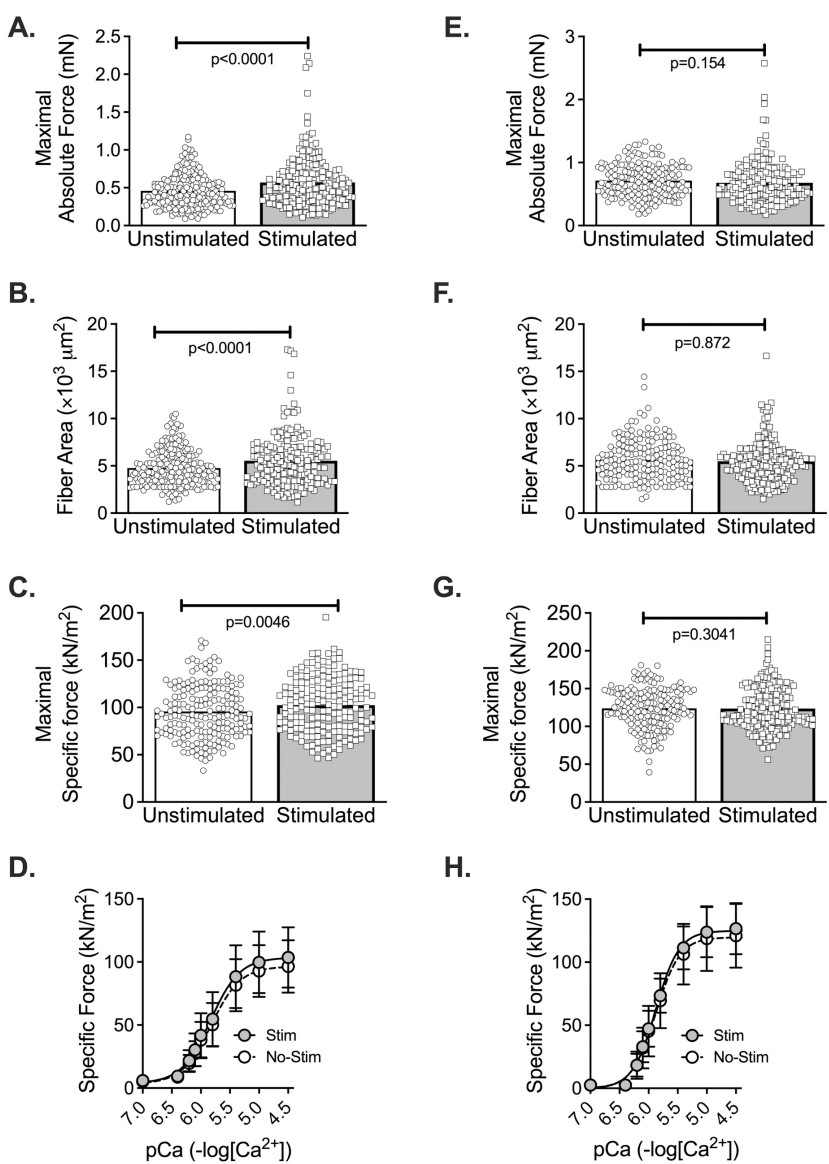

**Fig 2. Contractile properties of diaphragm slow and fast fibers.** In slow fibers, maximal calcium activated force (A), fiber cross-sectional area (B), and maximal specific force (C) were significantly greater in the stimulated hemidiaphragm. Averaged specific force vs pCa relationship (D). In fast fibers, no significant differences in maximal force (E), fiber cross-sectional area (F), maximal specific force (G), or the specific force vs pCa relationship (H) were detected. Mixed effect model; mean and standard deviation shown. A-C and E-G symbols are data from individual fibers. Statistical analysis by linear mixed model.

shows specific force vs pCa relationship for data averaged per subject. Fig S2 shows that there were no significant differences in calcium sensitivity (pCa50), rate-constant of force redevelopment (ktr, cross-bridge kinetics), and Hill coefficient (cooperativity). In fast fibers, we did not detect significant differences between stimulated and non-stimulated hemidiaphragms for any of the variables examined (Fig 2E–H and S2; also confirmed by non-parametric Mann-Whitney test; p = 0.413). We also examined the relationship between fiber area × force (Fig S3) and MV duration × specific force (Fig S4) in stimulated and non-stimulated hemidiaphragms. Fiber force was directly related to estimated cross-sectional area in slow (p < 0.001)

and fast fibers (p < 0.001) as expected, with a steeper slope for stimulated slow fibers (p = 0.001) that is consistent with higher specific force (Fig S3). Averaged specific force was inversely related to MV duration in unstimulated slow fibers (unstimulated fibers: p < 0.005, stimulated fibers: p = 0.153,) further supporting protection of VIDD with stimulation (Fig S4). There was no significant relationship between force and MV duration in fast fibers (stimulated fibers: p = 0.239, unstimulated fibers: p = 0.649).

## Protein abundance and post-translational modifications

There were no significant differences between sides in the relative abundance of the key structural, regulatory, and contractile proteins titin, nebulin, MyBP-C, MyHC, or actin. Similarly, no significant differences between stimulated and non-stimulated were evident in the abundance of titin or its degradation products (Fig 3), titin phosphorylation status (Fig 4), or abundance of titin-associated proteins (Fig 3). Total phosphorylation levels of nebulin and myosin binding protein C were also not different between sides (Fig 4). The protein calpain (CAPN3), which interacts with titin and is involved in sarcomere and cytoskeletal remodeling, has been implicated in VIDD. We measured total CAPN3 and CAPN3 degradation products (markers of activation) and found no difference between sides (Fig 3B). Similarly, the levels of the stress response proteins vinculin and CSRP3 (Fig 3C) were similar for each side.

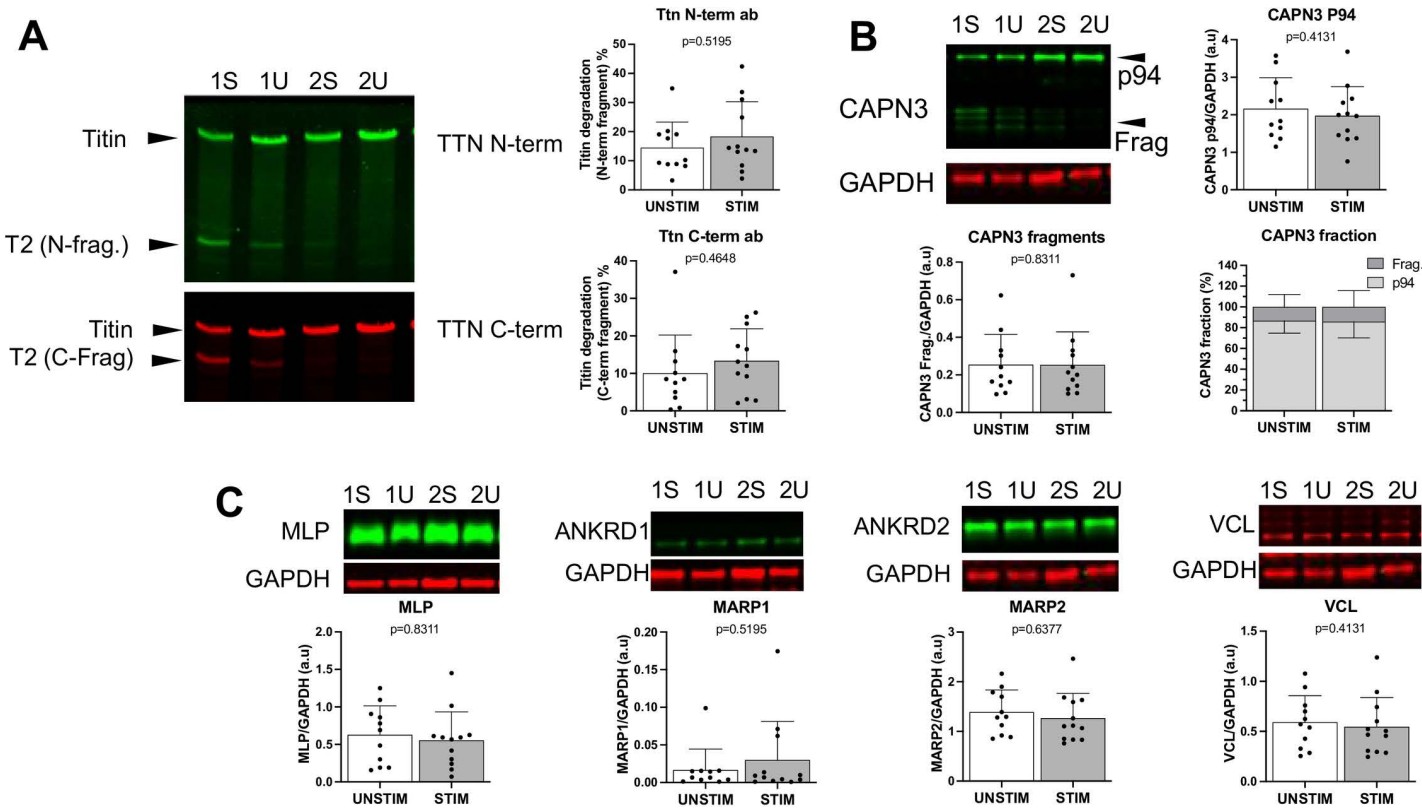

**Fig 3. Abundance of titin fragments and titin-binding proteins.** Western blot images and data of the N-terminal and C-terminal fragments of titin (A). (B) Calpain-3 (CAPN3), muscle specific protease at the N2A region of titin. Full-length, inactive calpain-3 (P94) and calpain fragments comprise the CAPN3 fraction. (C) Titin-binding proteins. Muscle Lim-Protein (MLP), telethonin-binding protein in the Z-disk region; muscle ankyrin repeat proteins 1 and 2 (MARP1 and MARP2), stress-responsive proteins at the N2A region of titin; and vinculin (VCL), membrane protein involved in sarcomere stability. Sample Western blot images are from the stimulated (STIM) and unstimulated (UNSTIM) sides of subjects 1 and 2. P-values shown from Wilcoxon tests; mean and standard deviation shown.

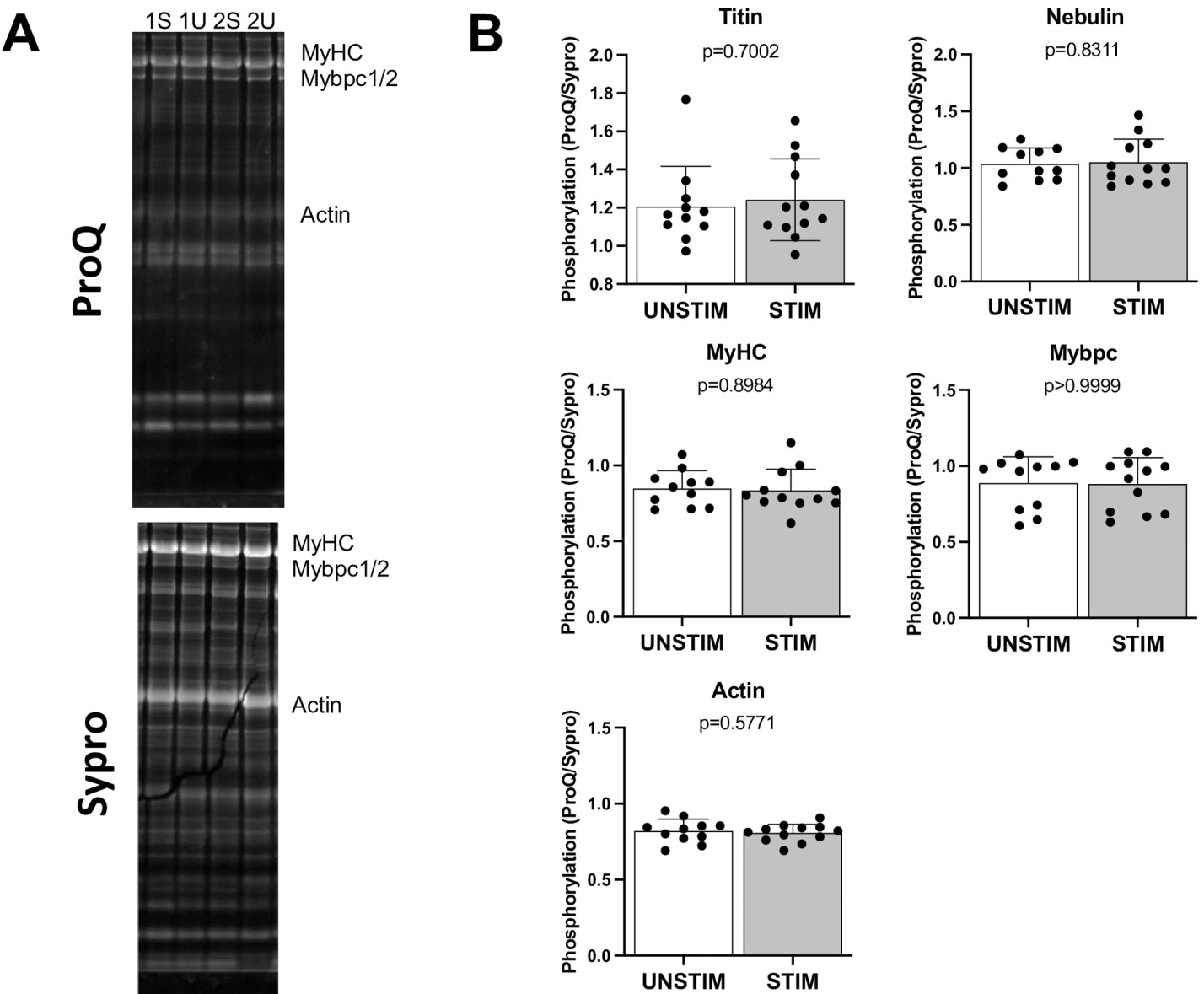

**Fig. 4. Sarcomere protein phosphorylation.** (A) Sample images of Pro-Q (phosphorylation) and SYPRO Ruby (total protein). Images are from the stimulated (STIM) and unstimulated (UNSTIM) sides of subjects 1 and 2. (B) Figures depict phosphorylation state of titin, nebulin, myosin heavy chain (MyHC), myosin binding protein **C** (Mybpc), and actin. P-values are from Wilcoxon tests; mean and standard deviation shown.

## Discussion

The primary novel findings of our study were that intermittent phrenic stimulation during 4–6 hours of open-chest thoracic surgery resulted in higher forces and estimated cross-sectional area in slow fibers. Higher absolute force and cross-sectional area were secondary analyses and results must be interpreted with caution. We also found an inverse relationship between non-stimulated slow fiber specific force (index of contractile function) and the time from intubation to biopsy, which tracked closely with duration of surgery in our study. Fast fibers were not affected by stimulation. Altogether the analyses of myofibrillar proteins, phosphorylation, titin-linked or stress-responsive proteins, and markers of protein synthesis

and degradation pathways revealed no differences between stimulated and unstimulated diaphragm in whole-tissue homogenates.

Diaphragm weakness in thoracic surgery: evidence and mechanisms Diaphragm fiber specific force decreases in slow and fast fibers with two hours of thoracic surgery whereas myofibrillar structure and fiber size are unchanged during this timeframe [1]. Our data reinforces the notion that thoracic surgery causes loss of diaphragm fiber force. Importantly, our study includes the largest cohort tested to our knowledge and focuses on patients without pulmonary dysfunction or pre-existing inspiratory muscle weakness. An additional novel insight is a progressive and exacerbated decline in slow fiber specific force as sugery duration exceeds 5 hours. The exact mechanisms for diaphragm weakness with thoracic surgery are unclear, but loss of skeletal muscle force is not widespread as thoracic surgery did not cause weakness in fibers of latissimus dorsi muscle. Overall, previous studies and our current findings suggest that while diaphragm fiber function deteriorates rapidly, systemic factors are not determinants of dysfunction during surgery.

The diaphragm is constantly active during spontaneous breathing and inactivity is a critical determinant of diaphragm fiber weakness [28,29]. Open-chest thoracic surgery requires mechanical ventilation (MV), and the consensus from animals and human studies is that MV impairs diaphragm fiber contractile function [6,30–32]. Although a previous study showed that MV in the absence of critical illness did not impair contractile function of permeabilized single fibers [33], during open chest surgery the diaphragm is fully quiescent and does not undergo the same mechanical forces as when MV occurs with a closed chest such as cyclic stretches that may exert protective effects agains dysfunction [29]. Moreover, the surgical procedures performed in our study and thoracic surgeries in general also require cardiopulmonary bypass, which can further impair diaphragm contractile function [34]. Serial ultrasounds identified post-operative diaphragm dysfunction in 38 of 100 patients, following elective cardiac surgeries [35]. While no preoperative patient attributes distinguished those who eventually developed diaphragm dysfunction, average cardiopulmonary bypass time was 87% longer in patients with postoperative diaphragm dysfunction. These observations are consistent with cardiopulmonary bypass exacerbating diaphragm dysfunction. The mechanisms of diaphragm dysfunction with mechanical ventilation and cardiopulmonary bypass remain unclear, with limited studies in humans. Decreased blood flow and hypoxia [36], impairments in neuromuscular transmission [34,37], mechanical unloading of titin [29,38], activation of proteolytic pathways and inactivation of protein synthesis [6,30,39], abnormal calcium release [40], and sarcomeric protein dysfunction [6,7,41] have all been implicated in the diaphragm dysfunction caused by mechanical ventilation and surgery.

## Clinical implications of inspiratory muscle dysfunction in thoracic surgery

Diaphragm weakness has implications for clinical management of patients recovering from thoracic surgery. Beyond maintenance alveolar ventilation, and perhaps of greater clinical relevance, normal diaphragm function is required for proper airway clearance – a forceful and rapid diaphragm contraction precedes the expiratory phase and is key determinant of an effective cough that minimizes postoperative pulmonary complications [42–44]. A recent large multinational study found that 55% of patients who underwent cardiac surgery with cardiopulmonary bypass experienced at least one post-operative pulmonary complication [45]. Pulmonary complicatons are the most common problem after thoracic surgery [3,46,47] and sonographic evidence of poor pre-operative diaphragm function is independently associated with post-operative pulmonary complications and increased ICU length of stay [3]. Post-operative diaphragm dysfunction resulted in more frequent pulmonary complications, leading to slower weaning from MV and longer ICU recovery time [35,48]. Among the risk factors

contribute to postoperative complications, intraoperative PEEP and neuromuscular blockade, which worsen existing diaphragm dysfunction [49,50]. Thus, diaphragm weakness may delay weaning of MV and prolong hospital stay not only in patients with critical illness but also those recovering from thoracic surgeries [35]. Therefore, it is important to target the mechanisms and advance pre- and perioperative therapies that minimize diaphragm dysfunction caused by thoracic surgery.

## Phrenic stimulation during mechanical ventilation and thoracic surgery

Inactivity is considered a determinant of diaphragm weakness caused by mechanical ventilation and, presumably, thoracic surgery. In this context, diaphragm activation via the phrenic nerve during mechanical ventilation and surgery is a logical approach to mitigate dysfunction. Recent studies suggest phrenic stimulation is a feasible, safe, and effective approach to enhance inspiratory pressure production and promote ventilator weaning in patients unable to complete voluntary inspiratory strengthening exercises [51]. Studies in animal models and humans have shown that phrenic stimulation during MV prevented or attenuated the decrease in diaphragm thickness, fiber cross-sectional area, and contractile function [10,11,13,52,53]. In comparison to our trial (1Hz stimulation for 60 seconds, every 30 minutes), these previous studies examined longer periods of MV (rats: 18 hours [10], pigs: 19–35 hours [11], humans: 48 to 60 hours [13,52], sheep: 72 hours [53]) and employed protocols with higher stimulus frequency (20–30 Hz) and train rates (every 20 seconds in rats [10], from every breath to every fourth breath in humans [13,52], and every breath in sheep [53]). Rat diaphragm specific force assessed in 'intact' bundles ex vivo was preserved when MV included phrenic stimulation, but fiber cross-sectional area was not examined [10] These previous studies of phrenic stimulation were performed during mechanical ventilation in a closed-chest scenario.

There have been limited investigations on perioperative phrenic stimulation during open-chest thoracic surgery, with no studies examining fast and slow diaphragm fiber contractile function. Our pilot study [14], which involved patients with pulmonary disorders and pre-existing inspiratory muscle weakness, suggested phrenic stimulation during thoracic surgery was feasible and had potential to attenuate loss of force in fast diaphragm fibers. In the current study, which included a larger cohort of patients with normal lung function and maximal inspiratory pressure, we observed a higher slow fiber maximal specific force and estimated fiber area in permeabilized fibers whereas fast fibers did not show statistically significant changes. The relationship between MV duration and slow fiber specific force suggests that the phrenic stimulation protocol we employed did not elicit full protection of slow fiber force.

Phrenic stimulation during MV in sheep attenuated the decrease in fast fiber cross-sectional area [53]. In the current study, slow, but not fast fiber cross-sectional area was greater with stimulation. This discrepancy may be related to differences in diaphragm mechanics during MV with open-chest, phrenic stimulation protocols, duration of MV, and species. In pigs, higher diaphragm thickness and fiber cross-sectional area were not accompanied by improved inspiratory pressure [11], and there was no functional assessment reported in humans with higher diaphragm thickness after stimulation during MV [13]. Since higher diaphragm thickness did not translate into better function in pigs [11], it is possible that phrenic stimulation during MV in larger mammals increases diaphragm thickness partially due to edema and the higher cross-sectional area may occur through mechanisms that do not preserve contractile function. However, we found that higher slow fiber cross-sectional area in the stimulated hemidiaphragm accompanied proportional increases in maximal fiber force (S4 Fig). The steeper slope for slow fiber area-to-maximal force relationship in the stimulated hemidiaphragm also reflects the

improvement in maximal specific force (Fig 2). The slope for fast fibers was also slightly steeper, but not statistically significant (P = 0.054), in the stimulated diaphragm. A different protocol might also elicit benefits to fast fibers. We must exercise caution comparing effects of phrenic stimulation during closed-chest MV vs open-chest MV during thoracic surgery. As mentioned above, the diaphragm is fully quiescent and dissociated from lung movement during open-chest procedures, which may exacerbate dysfunction induced by inactivity and poses a challenge for maintenance of function with sporadic phrenic stimulation. In general, our study reinforces the safety and feasibility of phrenic stimulation during thoracic surgery and advances the field by showing intraoperative feasibility and direct benefits to diaphragm fibers.

## Potential mechanisms of thoracic surgery and MV-induced diaphragm dysfunction and (partial) protection with phrenic stimulation

MV results in diminished or abolished drive to breathe with a potential for phrenic and neuromuscular transmission dysfunction [54]. Unloading the diaphragm triggers disuse atrophy and contractile dysfunction via abnormal proteostasis (increased protein degradation and decreased synthesis), impaired calcium release by the ryanodine receptor 1 [40,55], and post-translational modifications of sarcomeric proteins [56]. The permeabilized fiber preparation is void of all membranes and assesses sarcomeric protein function. Currently, there are no methods available to examine excitation-contraction coupling in diaphragm biopsies. The full protection of contractile dyfunction induced by MV in intact rodent diaphragm with phrenic stimulation is most likely due to combined improvements in excitation-contraction coupling and sarcomeric protein function. Phrenic stimulation may have protected excitation-contraction in our study but this is not possible to examine in diaphragm biopsies.

Titin-based mechanosensing has been proposed as a signaling hub for diaphragm dysfunction induced by unloading [29] Titin is a critical determinant of myofiber tension at optimal sarcomere length, and phrenic stimulation resulted in higher passive tension. However, there was no detectable effect of phrenic stimulation on titin abundance, degradation products, or titin-binding proteins that modulate stiffness (MARP1 and 2) in whole-tissue homogenates. The greater increase in slow fiber absolute force than specific force suggests that improvements elicited by phrenic stimulation were in proteostasis. Calpains have been implicated in heightened protein degradation and diminished protein synthesis, atrophy, and contractile dysfunction induced by MV [57]. Yet, we saw no effect of phrenic stimulation on abundance or markers of activation for any of those proteins. The exact mechanisms underlying sarcomeric protein dysfunction and loss of diaphragm specific force with MV and thoracic surgery remain unclear [28] The increase in maximal specific force without changes in calcium sensitivity suggests improvements in function of thick filament proteins. We assessed key thick-filament proteins and found no difference in their abundance or phosphorylation status.

Skeletal muscle unloading causes cross-bridges to enter a super-relaxed state that prevents interaction with actin and force generation [27,58]. One plausible scenario is that phrenic stimulation, by imposing intermittent load on the diaphragm during MV, mitigates the loss of specific force by minimizing the transition of cross-bridges to a super-relaxed state. We did not prepare the samples to assess cross-bridge states but this topic is worth investigating in future studies. Overall, we were unable to define potential molecular mechanisms associated with higher fiber area and force induced by phrenic stimulation in whole tissue homogenates. Fiber type-specific post-translational modification proteomics and synchrotron radiation x-ray diffraction will be necessary to resolve potential biochemical and biophysical events underlying the improvements in fiber size and force with phrenic stimulation. However, our study established that phrenic stimulation during thoracic surgery is feasible and has potential

to prevent or mitigate loss of diaphragm force and postoperative complications in patients with normal lung function and maximal inspiratory pressure.

### Limitations and methodological considerations

The main limitation of our study relates to number of subjects included from a subset of the clinical trial relying on a sample size estimated based another endpoint (published previously [59]). Based on the extensive experience of our group working with human samples, we expected the sample size would be enough to detect difference in the primary endpoint of specific force. We stimulated at twice the minimal threshold (up to 25 mA twitches) to achieve supramaximal stimulation to recruit all motor units and promote consistent physiological effect across patients. A submaximal activation would add variability due to differences in motor unit activation thresholds and physiological responses that depend on the ratio of submaximal-to-maximal activation. The twitch stimulations do not mimic the expected physiologic range of diaphragm contractions – firing frequency 4–10 Hz [60] – and were independent of lung inflation by MV. However, synchronization of phrenic stimulation and MV-delivered lung inflation is not necessary (or physiologically relevant) during open-chest thoracic surgery. The clinical environment did not permit us to make quantification of force generation, but we could clearly observe vigorous contractions of the stimulated side. The optimal parameters and timing of phrenic stimulation remains unresolved. Our goal was to test a protocol that was considered 'minimally disruptive' for the surgical procedures. However, the benefits we report here serve as impetus to optimize phrenic stimulation protocols and examine impact on duration of MV and post-operative pulmonary complications.

The selection of stimulated hemidiaphragm side based on surgeon discretion is not a statistical randomization. In this initial effort, we had to compromise the ideal experimental design to meet the surgical care of the patient. Moreover, supramaximal stimulation and contraction of a hemidiaphragm applies slight lengthening forces upon the contra-lateral unstimulated side. Diaphragm stretches might be enough to blunt the detrimental effects of MV and inactivity [61], and the unstimulated side may not have represented a pure "inactivity" control. In this case, our results would underestimate the benefits of intermittent phrenic stimulation. Finally, we measured protein abundance in whole-tissue homogenates to gain insight into potential mechanisms of protection. However, the effects with our protocol seem fiber type-specific, which would require single fiber analysis that are difficult to complete with the amount of tissue available for our study.

### Conclusion

Intermittent, unilateral phrenic stimulation during horacic surgery led to greater maximal specific force and fiber cross-sectional area in slow fibers of the stimulated hemidiaphragm, without significantly affecting fast diaphragm fiber function. Maximal specific force was positively associated with fiber cross sectional area and inversely related to MV duration, especially when surgeries exceed 5 hours duration. Abundance and post-translational modifications of myofibrillar proteins and titin-linked proteins from whole-tissue homogenates remained unchanged. These findings suggest that protection of maximal specific force occurred by a mechanism independent of effects on sarcomeric protein degradation. Our study supports further examination of intraoperative phrenic stimulation to optimize protocols for increases in fiber size or force and investigation of its potential to decrease postoperative complications.

### Supporting Information

**S1 Fig. Strontium sensitivity of myosin heavy chain (MyHC) isoforms determined by gel electrophoresis.** Slow and fast myosin heavy chain isoform classification was completed in 345 fibers from 10 subjects, using the ratio between force at pSr 4.0 and 5.0. A ratio of force at

pSr 5.0 to pSr 4.0 of 75% or greater indicated slow fibers, and ratio less than 75% indicated fast fibers. The pSr criteria were confirmed with MyHC separation by electrophoresis.
(TIF)

**S2 Fig. Passive and active mechanical properties of diaphragm fibers.** In slow diaphragm muscle fibers, there were no differences in calcium sensitivity (A), rate of tension development (B), or the Hill coefficient (C). However, maximal passive force (D) was significantly greater on the stimulated side of slow fibers. In fast diaphragm muscle fibers, no differences between the stimulated and unstimulated side were found for calcium sensitivity (**E**), rate of tension development (**F**), Hill coefficient (**G**), or passive force (**H**).
(TIF)

**S3 Fig. Relationship between fiber cross-sectional area and force.** In both slow (**A**) and fast (**B**) diaphragm muscle fibers, absolute fiber force was positively associated with the estimated cross-sectional area of the stimulated and unstimulated fibers.
(TIF)

**S4 Fig. Relationship between specific force and duration of mechanical ventilation.** In non-stimulated (NON-STIM) slow diaphragm fibers, mechanical ventilation (MV) duration was inversely correlated to the maximal specific force. The relationship between specific force and the duration of mechanical ventilation was not significant for fast diaphragm fibers or stimulated slow fibers.
(TIF)

**Table S1. Serious adverse events in study participants.**
(DOCX)

**Table S2. Nonserious adverse events in study participants.**
(DOCX)

**S1 File. Consort Checklist.**
(DOCX)

**S2 File. Study Protocol.**
(PDF)

**S3 File. Original Study Protocol Submission.**
(PDF)

**S4 File. Uncropped_Gels_Membrane-Images.**
(PDF)

**S1 Data. Individual single fiber data used for statistical analysis.**
(PDF)

**S2 Data. Individual single fiber pSr and gel data.**
(XLSX)

## Acknowledgements

We thank the Division of Cardiothoracic Surgery research coordinators with their regulatory assistance. We are grateful for the participation of the patients.

## Author contributions

**Conceptualization:** Thomas Beaver, A. Daniel Martin, Christiaan Leeuwenburgh, Coen A. C. Ottenheijm, Barbara K. Smith, Leonardo F. Ferreira.

**Data curation:** Guilherme Bresciani, Robbert van der Pijl, Robert Mankowski, Coen A. C. Ottenheijm, Shakeel Ahmed, Vinicius M. Mariani, Wei Xue, Barbara K. Smith, Leonardo F. Ferreira.

**Formal analysis:** Guilherme Bresciani, Thomas Beaver, A. Daniel Martin, Robbert van der Pijl, Robert Mankowski, Christiaan Leeuwenburgh, Coen A. C. Ottenheijm, Shakeel Ahmed, Vinicius M. Mariani, Wei Xue, Barbara K. Smith, Leonardo F. Ferreira.

**Funding acquisition:** Thomas Beaver, A. Daniel Martin, Barbara K. Smith.

**Investigation:** Guilherme Bresciani, Thomas Beaver, A. Daniel Martin, Robert Mankowski, Coen A. C. Ottenheijm, Tomas Martin, George Arnaoutakis, Shakeel Ahmed, Vinicius M. Mariani, Wei Xue, Barbara K. Smith, Leonardo F. Ferreira.

**Methodology:** Guilherme Bresciani, Thomas Beaver, A. Daniel Martin, Robbert van der Pijl, Robert Mankowski, Christiaan Leeuwenburgh, Coen A. C. Ottenheijm, Tomas Martin, George Arnaoutakis, Shakeel Ahmed, Vinicius M. Mariani, Wei Xue, Barbara K. Smith, Leonardo F. Ferreira.

**Project administration:** Thomas Beaver, A. Daniel Martin, Coen A. C. Ottenheijm, Barbara K. Smith, Leonardo F. Ferreira.

**Resources:** A. Daniel Martin, Christiaan Leeuwenburgh, Coen A. C. Ottenheijm, George Arnaoutakis, Barbara K. Smith, Leonardo F. Ferreira.

**Supervision:** Thomas Beaver, A. Daniel Martin, Christiaan Leeuwenburgh, Coen A. C. Ottenheijm, Tomas Martin, George Arnaoutakis, Barbara K. Smith, Leonardo F. Ferreira.

**Visualization:** Barbara K. Smith.

**Writing – original draft:** Guilherme Bresciani, Robbert van der Pijl, Wei Xue, Barbara K. Smith, Leonardo F. Ferreira.

**Writing – review & editing:** Guilherme Bresciani, Thomas Beaver, A. Daniel Martin, Robbert van der Pijl, Robert Mankowski, Christiaan Leeuwenburgh, Coen A. C. Ottenheijm, Tomas Martin, George Arnaoutakis, Shakeel Ahmed, Vinicius M. Mariani, Wei Xue, Barbara K. Smith, Leonardo F. Ferreira.

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
