## [Decision Letter · Decision Letter 0]

22 Aug 2024

PONE-D-24-20012Intraoperative phrenic nerve stimulation to prevent diaphragm fiber weakness during thoracic surgeryPLOS ONE

Dear Dr. Ferreira,

Thank you for submitting your manuscript to PLOS ONE. After careful consideration, we feel that it has merit but does not fully meet PLOS ONE’s publication criteria as it currently stands. Therefore, we invite you to submit a revised version of the manuscript that addresses the points raised during the review process.

We look forward to receiving your revised manuscript.

Kind regards,

Steven E. Wolf, MD

Academic Editor

PLOS ONE

Journal Requirements:

"NIH R01 AR072328, HL130318, HL121500"

3. Please expand the acronym “NIH R01” (as indicated in your financial disclosure) so that it states the name of your funders in full.

"We thank the Division of Cardiothoracic Surgery research coordinators with their regulatory assistance. We are grateful for the participation of the patients. The study was funded by NIH R01 AR072328. LFF was supported by HL130318 and CAC was supported by HL121500."

"NIH R01 AR072328, HL130318, HL121500"

Reviewers' comments:

Reviewer's Responses to Questions

**Comments to the Author**

1. Is the manuscript technically sound, and do the data support the conclusions?

Reviewer #1: Yes

Reviewer #2: Yes

2. Has the statistical analysis been performed appropriately and rigorously? 

Reviewer #1: No

Reviewer #2: Yes

3. Have the authors made all data underlying the findings in their manuscript fully available?

Reviewer #1: Yes

Reviewer #2: No

4. Is the manuscript presented in an intelligible fashion and written in standard English?

Reviewer #1: Yes

Reviewer #2: Yes

5. Review Comments to the Author

Reviewer #1: This is a report of a small cohort study. There are some deficiencies in the description of the statistical considerations of the study:

1. Sample size discussion is inadequate. What is the primary outcome of the study? What test statistic is being used? It appears that random effects models are being used in the analysis, yet there is no discussion of assumptions on the model that are necessary for computing sample size. Referencing a prior study is insufficient, as this study needs to stand alone. The Discussion should also include a sentence on whether the assumptions used in the sample size computation (variability etc.) were actually realized in the clinical trial.

2. Random effects models have assumptions that require validation of those assumptions. What diagnostics did you use to ensure the models are apt?

3. p<0.05 to declare significance is fine for the primary outcome, but there are other tests being done and there does not appear to be any adjustment for multiple testing. Of course the other tests may be exploratory, in which case p<0.05 really has no meaning.

Reviewer #2: GENERAL COMMENTS

Bresciani et al assessed the effects of unilateral phrenic nerve stimulation during cardiothoracic surgery on diaphragm muscle fiber function. This is huge work, with a very complex population and very complicated collection of diaphragm biopsies. I provided point-by-point comments below. Some of them are already described by the authors as limitations. However, I consider them to be very important points that need to be considered while reviewing/considering the manuscript for publication. I hope these comments help you improve the quality of the manuscript.

There some differences between the clinical trials registration and the results reported here.

ABSTRACT

Please, clarify details of the stimulation (e.g., duration of each stimulation and etc)?

MAIN TEXT

It is not clear why the focus of this study was patients with normal lung function/inspiratory muscle strength. The question would be much more relevant in high-risk populations, such as patients with lung diseases.

The study used unilateral phrenic nerve stimulation and considered the other side as an internal control. Unilateral phrenic nerve stimulation can sometimes cause both hemidiaphragms to contract. This phenomenon occurs due to the presence of cross-innervation or central connections between the motor neurons controlling the diaphragm. When one phrenic nerve is stimulated, it can lead to a bilateral response, causing both hemidiaphragms to contract, although the effect is typically stronger on the side directly stimulated it imposes a limitation on what is considered the “control hemidiaphragm” for the study. This may underestimate the effect of the intervention, which could affect the conclusions that, for instance, only one type of fiber was found to be different.

Please, clarify whether or not neuromuscular block was used at any point in any patients for the surgical procedure.

If the protocol was extended to include patients with lung transplant, why reporting partial data only here? and from patients with preserved MIP? The results from patients post-lung transplant would be much more interesting from a clinical perspective.

Line 164. I am not sure I understand the parameters for the stimulation correctly. Please, clarify: each stimulation was continuously applied for 1 minute? If that is correct, I do not understand the rationale for these settings. Each inspiratory effort last approximately 0.8 to 1.2 seconds. I would expect a stimulation targeted to protect the diaphragm to have such a physiological range. If the stimulation lasted one minute, this does not resemble the way the diaphragm works normally. Or did the stimulation last 1.5 ms? If this is the case, this is the complete opposite and similarly does not resemble how the diaphragm works normally. Please, clarify the parameters in the text. I suggest this is addressed in the limitations section and please describe the rationale for this choice.

Sample size was based on prior work about mitochondrial function. The effect size or the magnitude of the difference between the two groups used for the sample size calculation is not described. Moreover, mitochondrial function is not a primary outcome in the study. Please, clarify the rationale for this approach and why not calculate the sample size based on differences expected for the primary outcomes of the study?

Calculated sample size was n=20 but 25 patients were included. Please, justify. Then, data is available from 19 patients, which is less than the calculated sample size.

Looking at the first clinical trial registration (https://clinicaltrials.gov/study/NCT03303040?tab=history&a=1#version-content-panel), estimated enrollment was 54 and not all outcomes described in the clinical trials registration are reported here. Please, clarify.

How was the intensity of the stimulation set since it was not the same for all? Based on the strength of the diaphragm contraction? The effect of the stimulation intensity and the effort generated by the diaphragm during each contraction caused by the stimulation could significantly affect the study results.

The authors discuss that “However, synchronization of phrenic stimulation and MV-delivered lung inflation is not necessary (or physiologically relevant) during open-chest thoracic surgery. The clinical environment did not permit us to make quantification of force generation, but we could clearly observe vigorous contractions of the stimulated side. The optimal parameters and timing of phrenic stimulation remains unresolved.” However, studies have shown that, during mechanical ventilation, the timing (i.e., synchronous or dyssynchronous) type of contraction (e.g., concentric vs eccentric) and the effort generated by the diaphragm (e.g., large vs small effort) all can affect whether it will develop injury or not. These factors can influence diaphragm function independently or in combination.

6. PLOS authors have the option to publish the peer review history of their article (what does this mean? ). If published, this will include your full peer review and any attached files.

**Do you want your identity to be public for this peer review?** For information about this choice, including consent withdrawal, please see our Privacy Policy .

Reviewer #1: No

Reviewer #2: No

---

## [Author Response · Author response to Decision Letter 1]

16 Jan 2025

We thank the reviewers and editor for providing comments to improve our manuscript. We have made all changes requested and responded to the comments. The point-by-point response is included as a separate file.

---

## [Decision Letter · Decision Letter 1]

27 Feb 2025

Intraoperative phrenic nerve stimulation to prevent diaphragm fiber weakness during thoracic surgery

PONE-D-24-20012R1

Dear Dr. Ferreira,

We’re pleased to inform you that your manuscript has been judged scientifically suitable for publication and will be formally accepted for publication once it meets all outstanding technical requirements.

Kind regards,

Steven E. Wolf, MD

Academic Editor

PLOS ONE

Additional Editor Comments (optional):

Reviewers' comments:

Reviewer's Responses to Questions

**Comments to the Author**

1. If the authors have adequately addressed your comments raised in a previous round of review and you feel that this manuscript is now acceptable for publication, you may indicate that here to bypass the “Comments to the Author” section, enter your conflict of interest statement in the “Confidential to Editor” section, and submit your "Accept" recommendation.

Reviewer #1: All comments have been addressed

2. Is the manuscript technically sound, and do the data support the conclusions?

Reviewer #1: (No Response)

3. Has the statistical analysis been performed appropriately and rigorously? 

Reviewer #1: (No Response)

4. Have the authors made all data underlying the findings in their manuscript fully available?

Reviewer #1: (No Response)

5. Is the manuscript presented in an intelligible fashion and written in standard English?

Reviewer #1: (No Response)

6. Review Comments to the Author

Reviewer #1: (No Response)

7. PLOS authors have the option to publish the peer review history of their article (what does this mean? ). If published, this will include your full peer review and any attached files.

**Do you want your identity to be public for this peer review?** For information about this choice, including consent withdrawal, please see our Privacy Policy .

Reviewer #1: No

---

## [Editor Report · Acceptance letter]

PONE-D-24-20012R1

PLOS ONE

Dear Dr. Ferreira,

I'm pleased to inform you that your manuscript has been deemed suitable for publication in PLOS ONE. Congratulations! Your manuscript is now being handed over to our production team.

Kind regards,

on behalf of

Dr. Steven E. Wolf

Academic Editor

PLOS ONE